Methods

# Nanobodies as novel tools to monitor the mitochondrial fission factor Drp1

Theresa Froehlich[1] , Andreas Jenner[2] , Claudia Cavarischia-Rega[3], Funmilayo O Fagbadebo[1], Yannic Lurz[4], Desiree I Frecot[1], Philipp D Kaiser[5], Stefan Nueske[6] , Armin M Scholz[6] , Erik Schäffer[4], Ana J Garcia-Saez[2,7], Boris Macek[3], Ulrich Rothbauer[1,8]

**In cells, mitochondria undergo constant fusion and fission. An essential factor for fission is the mammalian dynamin-related protein 1 (Drp1). Dysregulation of Drp1 is associated with neurodegenerative diseases including Parkinson's, cardiovascular diseases and cancer, making Drp1 a pivotal biomarker for monitoring mitochondrial status and potential pathophysiological conditions. Here, we developed nanobodies (Nbs) as versatile binding molecules for proteomics, advanced microscopy and live cell imaging of Drp1. To specifically enrich endogenous Drp1 with interacting proteins for proteomics, we functionalized high-affinity Nbs into advanced capture matrices. Furthermore, we detected Drp1 by bivalent Nbs combined with site-directed fluorophore labelling in super-resolution STORM microscopy. For real-time imaging of Drp1, we intracellularly expressed fluorescently labelled Nbs, so-called chromobodies (Cbs). To improve the signal-to-noise ratio, we further converted Cbs into a "turnover-accelerated" format. With these imaging probes, we visualized the dynamics of endogenous Drp1 upon compound-induced mitochondrial fission in living cells. Considering the wide range of research applications, the presented Nb toolset will open up new possibilities for advanced functional studies of Drp1 in disease-relevant models.**

## Introduction

Mitochondrial morphology is controlled by the balance between two opposing processes, fusion and fission. These highly regulated events adapt the cellular mitochondrial network to the bioenergetic requirements of the cell and maintain the functional integrity of mitochondria in important cellular processes such as iron metabolism, lipid biosynthesis, calcium homeostasis and cell death (Vantaggiato et al, 2019; Iwata et al, 2020; Rasmussen et al, 2020). Mitochondrial fission in higher vertebrates is primarily mediated by the large GTPase dynamin-related protein 1 (*DNM1L*; Drp1 also known as DLP1). The fission process follows a coordinated sequence of events in which cytosolic Drp1 is initially recruited to mitochondrial contact sites with the ER, facilitated by ER membrane-associated actin filaments (Li et al, 2015; Ji et al, 2017; Chakrabarti et al, 2018). On the mitochondrial outer membrane (MOM), Drp1 associates with adaptor proteins including the mitochondrial dynamics protein of 49 kD (MiD49, also known as MIEF2), mitochondrial dynamics protein of 51 kD (MiD51; also known as MIEF1), and presumably the mitochondrial fission factor (Mff) and fission protein 1 (FIS1) (Loson et al, 2013; Palmer et al, 2013; Osellame et al, 2016). Subsequently, Drp1 oligomerizes into ring-like or helical structures surrounding mitochondria and GTP hydrolysis then triggers a conformational change in Drp1 causing the contractive division of the mitochondrial network (De Vos et al, 2005; Friedman et al, 2011; Frohlich et al, 2013; Korobova et al, 2013; Bai et al, 2015). In addition to mitochondria, Drp1 also mediates the division of peroxisomes in combination with the adaptor proteins Mff and FIS1 (Li & Gould, 2002; Koch et al, 2003; Koch & Brocard, 2012).

Dysregulation of Drp1 is associated with neurodegenerative diseases such as Alzheimer's (Cho et al, 2009; Wang et al, 2009), Huntington's (Song et al, 2011; Shirendeb et al, 2012), or Parkinson's (Han et al, 2020) as well as cardiovascular diseases (Nan et al, 2017; Jin et al, 2021). For example, immunoblotting and immunocytochemical analyses revealed reduced Drp1 protein levels in hippocampal tissues of Alzheimer's patients compared with age-matched healthy individuals (Wang et al, 2009), whereas Huntington's disease patients were found to have increased Drp1 enzymatic activity mediated by interaction with mutant Huntingtin (Htt) aggregates, resulting in abnormal mitochondrial dynamics, including defective anterograde mitochondrial movement and

---

[1]Pharmaceutical Biotechnology, Eberhard Karls University Tübingen, Tübingen, Germany [2]Institute for Genetics and Cologne Excellence Cluster on Cellular Stress Responses in Aging-Associated Diseases (CECAD), University of Cologne, Cologne, Germany [3]Quantitative Proteomics, Department of Biology, Institute of Cell Biology, Eberhard Karls University Tübingen, Tübingen, Germany [4]Center for Plant Molecular Biology (ZMBP), Eberhard Karls University Tübingen, Tübingen, Germany [5]NMI Natural and Medical Sciences Institute at the University of Tübingen, Reutlingen, Germany [6]Livestock Center of the Faculty of Veterinary Medicine, Ludwig Maximilians University Munich, Munich, Germany [7]Max Planck Institute of Biophysics, Frankfurt, Germany [8]Cluster of Excellence iFIT (EXC2180) "Image-Guided and Functionally Instructed Tumor Therapies," University of Tübingen, Tübingen, Germany

Correspondence: ulrich.rothbauer@uni-tuebingen.de

synaptic deficiencies (Song et al, 2011; Shirendeb et al, 2012). More recently immunoblotting revealed lower phosphorylation levels of Drp1 at serine 616 (S616) in dermal fibroblasts from Parkinson's patients compared with healthy individuals (Han et al, 2020). In addition, elevated Drp1 expression and phosphorylation on S616 detected by immunoblotting and immunocytochemical analyses have been linked to cancer, including melanoma, glioblastoma, lung, breast, thyroid, pancreatic, and head and neck cancer (Rehman et al, 2012; Zhao et al, 2013; Ferreira-da-Silva et al, 2015; Kashatus et al, 2015; Serasinghe et al, 2015; Liang et al, 2020; Huang et al, 2022). It is assumed that an increased Drp1 level promotes mitochondrial cleavage and thus aerobic glycolysis, thereby promoting the growth and metastasis of tumor cells (Liang et al, 2020). In this context, knockdown of Drp1 was shown to be associated with a marked reduction in cancer cell proliferation and an increase in spontaneous apoptosis (Rehman et al, 2012; Kashatus et al, 2015; Liang et al, 2020). These findings highlight the growing importance of Drp1 as a biomarker or potential target for therapeutic interventions. However, despite several in vitro and in vivo studies, detailed information on the molecular mechanism and structural changes of Drp1 during mitochondrial fission, as well as its cellular dynamics and interacting components, is still lacking (Giacomello et al, 2020; Tong et al, 2020; Zerihun et al, 2023). This gap in knowledge is partly because of the limited availability of research tools to study Drp1. Notably, most live cell analyses rely on ectopic expression of fluorescent fusion constructs or epitope-tagged Drp1 (Labrousse et al, 1999; Solesio et al, 2013; Ji et al, 2017; Michalska et al, 2018; Xiong et al, 2022). However, recent studies revealed that N- or C-terminal labelling of Drp1 with fluorescent proteins or epitope tags leads to altered oligomerization dynamics and impairs its GTPase activity (Montecinos-Franjola et al, 2020).

Single-domain antibody fragments derived from the heavy chain-only antibodies of camelids (Hamers-Casterman et al, 1993), also known as nanobodies (Nbs), have emerged as versatile research tools in biomedical research. Because of their specific binding properties, small size, high stability, and solubility, these binders became an attractive alternative to conventional antibodies in many biochemical and cell biological research applications (Frecot et al, 2023). In addition, intracellular functional Nbs genetically fused to fluorescent proteins, so-called chromobodies (Cbs), serve as imaging probes to visualize dynamic changes of target antigens without genetic or covalent modifications and can be applied in different cellular compartments as well as in whole organisms (Wagner & Rothbauer, 2020). Here, we have identified a set of novel Nbs specifically recognizing human Drp1. After their detailed biochemical, biophysical and functional characterization, we have developed them as broadly applicable research tools. Thus, we converted them into affinity matrices for proteomic analysis of Drp1 and developed bivalent Nbs as labelling probes to detect endogenous Drp1 in confocal and super-resolution microscopy (SRM). By engineering turnover-accelerated Drp1-Cbs, which have an improved signal-to-noise ratio, we were able to visualize the drug-induced recruitment of Drp1 to mitochondria in living cells. Considering the broad spectrum of applications, we propose that the Drp1-specific Nbs/Cbs described herein open up new possibilities for future advanced functional studies of Drp1 in disease-relevant models and represent a promising alternative to currently available approaches.

## Results

### Identification of Drp1-specific Nbs

To generate Nbs against human Drp1 (Drp1), we immunized an alpaca (*Vicugna pacos*) with recombinant Drp1 using a 91-d immunization protocol and detected a specific immune response in a serum ELISA 63 d after the first vaccination (Fig S1A). Subsequently, we established an Nb phagemid library (size ~2 × 10$^7$ clones) from the mRNA of the peripheral B lymphocytes, which was subjected to phage display using either passively adsorbed or biotinylated Drp1. After two rounds of biopanning, we analysed 260 individual clones in a phage ELISA and identified 27 positive binders (Fig S1B) with eight unique sequences (Table S1). Notably, only two Nbs, D7 and D63, displaying highly diverse complementarity determining regions 3 (Fig 1A), showed sufficient binding signals in a protein ELISA using periplasmic extracts derived from Nb-expressing *Escherichia coli* (*E. coli*) (Fig S1C). Therefore, we purified these binders by immobilized metal affinity chromatography and subsequent size exclusion chromatography (Fig 1B). For determining their binding affinities, both Nbs were subjected to biolayer interferometry. Biotinylated Nbs were immobilized on streptavidin (SA) biosensors and binding kinetics were measured by titrating varying concentrations of Drp1. Both Nbs, D7 and D63, showed affinities in the low nanomolar range with $K_D$ values of ~3.5 and ~1.8 nM, respectively (Figs 1C and S2). In addition, analysing their folding stability using nano-differential scanning fluorimetry (NanoDSF) revealed that both Nbs exhibit stable folding and a low tendency to aggregate having melting temperatures of ~68 and ~70°C for D7 and D63, respectively (Fig 1D). To further assess whether Nb binding affects the functionality of Drp1, we measured the GTPase activity of Drp1 in the presence of the respective Nbs in vitro. Whereas D7 and the negative control (GFP Nb) had no effect on the enzymatic activity of Drp1, we observed a significant increase in the GTPase rate by a factor of ~1.7 after addition of a 10-fold molar excess of D63 (Fig 1E).

### Selected Drp1-Nbs recognize different domains of Drp1 and precipitate endogenous Drp1

Considering that Drp1 is composed of multiple domains including an N-terminal GTPase (aa 1–337), an unfolded (middle, stalk, aa 338–502), a variable (B insert, aa 503–635), and a C-terminal GTPase effector domain (GED) region (aa 636–736) (Otera et al, 2013), we constructed a series of GFP-labelled domain deletions of Drp1 (Fig 2A) and tested whether the selected Nbs recognize specific domains within Drp1 by pull-down assays. To this end, we generated so-called Drp1 nanotraps, for which we covalently immobilized D7 and D63 on N-hydroxysuccinimide (NHS)-activated sepharose beads via primary amino groups of accessible lysine residues (Traenkle et al, 2020; Fagbadebo et al, 2022). For binding analysis, the Drp1 nanotraps were incubated with soluble protein fractions of HEK293 cells transiently expressing the Drp1 deletion constructs.

https://doi.org/10.26508/lsa.202402608    vol 7 | no 8 | e202402608    

**Figure 1. Identification and biochemical characterization of Drp1-specific nanobodies.**
**(A)** Amino acid (aa) sequences of the complementary determining region 3 (CDR3) of the nanobodies (Nbs) D7 and D63 identified by protein ELISA. **(B)** Recombinant expression and purification of D7 and D63 by immobilized metal affinity chromatography and size exclusion chromatography. Coomassie staining of purified Nbs (2 μg) is shown. **(C)** Biolayer interferometry-based affinity measurements exemplarily shown for D7. Biotinylated Nb D7 was immobilized on streptavidin sensors. Kinetic measurements were performed using four concentrations of purified Drp1 ranging from 15.6–125 nM (displayed with gradually darker shades of color). The binding affinity ($K_D$) was calculated from global 1:1 fit shown as dashed lines. Affinities ($K_D$), association constants ($k_{on}$), and dissociation constants ($k_{off}$) of D7 and D63 determined by BLI shown as mean ± SD. **(D)** Stability analysis by nano-differential scanning fluorimetry displaying the fluorescence ratio (350/330 nm) first derivative for D7 (orange) and D63 (blue). Data are shown as the mean value of three technical replicates. **(E)** GTP hydrolysis of 100 nM Drp1 in solution in the presence of increasing concentrations of different Nbs at 37°C (mean ± SED, n = 5 experiments on 4 d). Significance was tested using two-way ANOVA and multiple comparison analysis in GraphPad Prism (n.s. $P > 0.05$, *$P < 0.05$, ****$P < 0.0001$).

Here we used the GFP-Trap as a positive control (Rothbauer et al, 2008). Immunoblot analysis revealed that both Nbs precipitated full-length Drp1 as expected (Fig 2B). However, whereas D7 exclusively recognizes the N-terminal GTPase domain, D63 only binds in the presence of the GED, whereas none of the Nbs binds nonspecifically to GFP (Fig 2B). Next, we tested the capacity of the Drp1 nanotraps to capture endogenous Drp1. Therefore, we generated soluble protein extracts from three different human cell lines (HEK293, HeLa, and U2OS) and incubated them with the Drp1 nanotraps. For comparison, we used a monoclonal anti-Drp1 antibody immobilized on proteinA/G-sepharose as positive control (PC) and the GFP-Trap as negative control (NC). Immunoblot analysis of all cell lines tested showed that both Drp1 nanotraps precipitated higher levels of endogenous Drp1 compared with the conventional antibody (Figs 3A and S3A and B).

For a more detailed insight, we analysed the proteome of both Drp1 nanotraps by mass spectrometry (MS) derived from the soluble protein fractions of HEK293, HeLa and U2OS cells. Immunoprecipitations were performed for all cell lines in three replicates, each with the same number of cells as starting material. As exemplarily shown for HEK293 cells, the results were highly reproducible showing a Spearman rank correlations close to one (Fig S4A). As expected, the correlation between the two different Drp1 nanotraps was higher than between each nanotrap and the control (GFP-Trap), confirmed by principal component analysis (Fig S4B). We next evaluated how well each nanotrap captured endogenous Drp1. Both nanotraps enabled the identification of multiple Drp1 peptides comparably, resulting in high coverage of the Drp1 sequence (~70%) (Fig S4C). Notably, 55 "Razor" peptides matching Drp1 isoform 1, 5, 6, or 7 were found in the precipitates of both nanotraps (Fig S4D). However, based on our dataset, we could not distinguish between these isoforms. Next, we confirmed the efficacy of Drp1 enrichment. Overall, the number of protein groups (PGs) identified was comparable for all samples, with a background of up to 981 proteins. For both nanotraps, Drp1 was one of the most abundant proteins detected, whereas it was not identified in the negative control (Fig 3B). Furthermore, Drp1 was the most significantly enriched protein with both traps (Fig 3C and D). Finally, we scanned

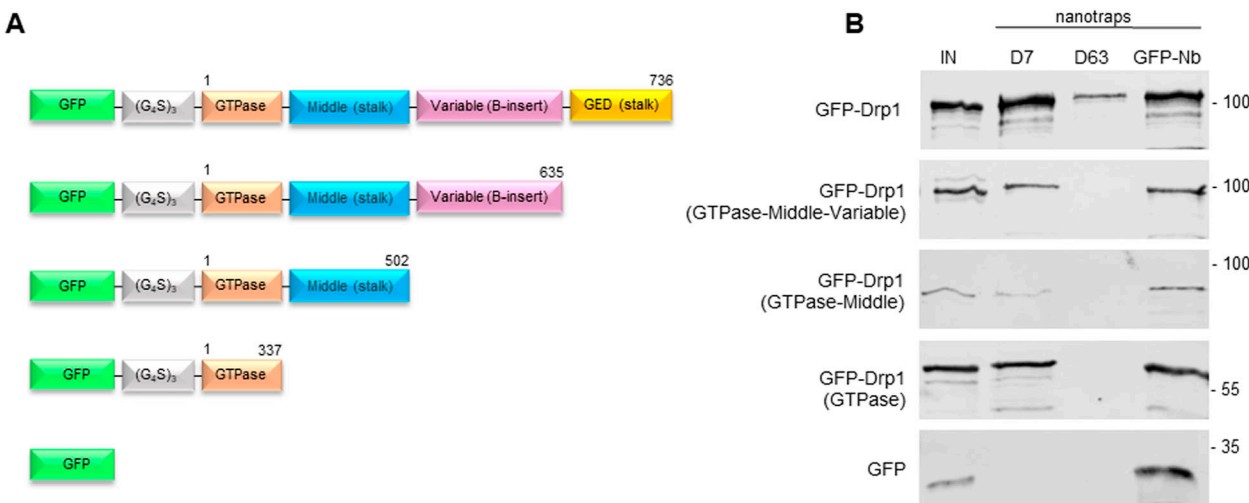

**Figure 2. Domain mapping of Drp1-Nbs.**
**(A)** Schematic illustration of the GFP-labelled Drp1 deletion constructs used for domain binding studies. Numbers indicate amino acid positions of the Drp1 coding sequence. **(B)** Results of the pulldown analysis of the individual Drp1 deletions as depicted in (A) using the Drp1-specific nanotraps (D7, D63) or the GFP-Trap (GFP-Nb) as positive control. Input (IN, 1% of total) and bound (20% of total) fractions were subjected to SDS–PAGE followed by immunoblot analysis using an anti-GFP antibody. Molecular weights in kD are indicated on the right.

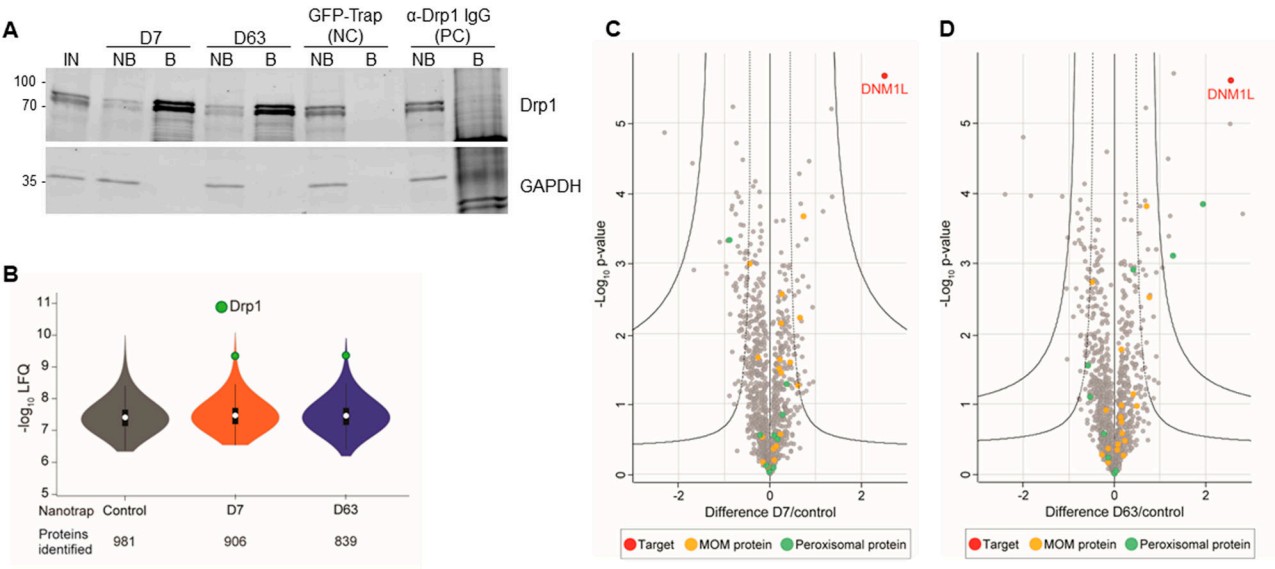

**Figure 3. Immunoprecipitation of endogenous Drp1 derived from HEK293 cells with mass spectrometry analysis of the interactome of the Drp1 nanotraps.**
**(A)** Representative result of an immunoprecipitation of endogenous Drp1 with Drp1-specific nanotraps (D7 and D63) using soluble protein fraction of HEK293 cells. As negative control (NC), the GFP-Trap as non-specific nanotrap was used and bead-coupled anti-Drp1 IgG served as a positive control (PC). Input (IN, 1% of total), non-bound (NB, 1% of total) and bound (B, 33% of total) fractions were subjected to SDS–PAGE followed by immunoblot analysis using antibodies specific for Drp1 (upper panel) and GAPDH (lower panel). Molecular weights in kD are indicated on the left. **(B)** Violin plot of the averaged label-free quantification intensities of the proteins identified from immunoprecipitation using HEK cells. White circles show the median; box limits indicate the 25th and 75th percentiles as determined by R software; whiskers extend 1.5 times the interquartile range from the 25th and 75th percentiles; polygons represent density estimates of data and extend to extreme values. Drp1 is marked in green. **(C, D)** Multi-volcano analysis (Hawaiian plot) of the pull-down of D7 (C) and D63 (D) against the negative control. $\log_2$ transformed ratios of the difference between the Drp1 nanotrap and the control (x axis) are plotted against the $\log_{10}$ transformed $P$-value (y axis). Drp1 is marked in red, mitochondrial outer membrane proteins are marked in yellow and peroxisomal proteins in green. Significant interactors can be class A hits (higher confidence, $s_0 = 0.1$, FDR = 0.01) or class B hits (lower confidence, $s_0 = 0.1$, FDR = 0.05), thresholds are displayed as a solid line and a dashed line, respectively.

our MS data for potential interaction partners of Drp1. Interestingly, both Drp1 nanotraps facilitated the enrichment of cytoplasmic or mitochondrial interactors compared with the control, whereas only the D63 nanotraps allowed for the enrichment of class A interactors

of peroxisomal proteins such as HSDL2 (hydroxysteroid dehydrogenase-like protein 2) or HSD17B4 (peroxisomal multifunctional enzyme type 2) (Fig 3C and D; Tables S2 and S3). In addition, with both Drp1 nanotraps, we found a significant

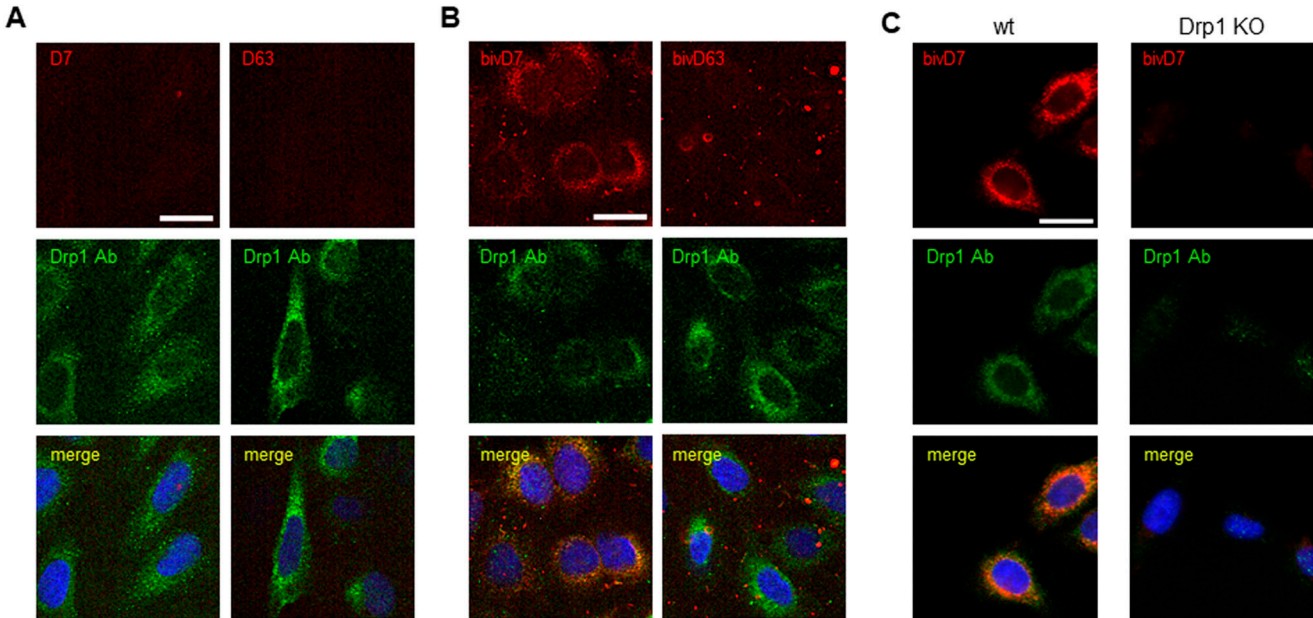

**Figure 4. Immunofluorescence staining with Drp1-Nbs.**
**(A)** Representative confocal laser scanning microscopy images of U2OS cells stained with D7 and D63 Nbs as primary labelling probes detected with an anti-VHH antibody labelled with Cy5. As positive control, cells were co-stained with anti-Drp1 antibody (Drp1 Ab) followed by detection with an AlexaFluor488-labelled secondary antibody. **(B)** Immunofluorescence (IF) detection of Drp1 in fixed and permeabilized U2OS cells after staining with bivalent versions of D7 and D63 (bivD7, bivD63) as described in (A). **(C)** Immunofluorescence (IF) detection of Drp1 in fixed and permeabilized wt HeLa (left column) and Drp1 KO HeLa cells (right column) stained with bivD7 as described in (B). **(A, B, C)** Nuclei were counterstained with DAPI. Scale bar 25 μm.

enrichment of CDGSH iron-sulfur domain-containing protein 1 (CISD1) and Histone deacetylase 6 (HDAC6), two factors previously reported to interact with Drp1 (English & Barton, 2021; Hua et al, 2021) (Tables S2 and S3).

Notably, both Drp1 nanotraps performed similarly in immunoprecipitation of Drp1 when using HeLa and U2OS cells and proteomic analysis of the bound fraction yielded comparable results (Figs S5A–E and S6A–E). Interestingly, we observed an additional enrichment of isoform 2 of Drp1 in HeLa cells with both nanotraps (Fig S5F and G), whereas in U2OS cells, both nanotraps also precipitated isoforms 4 or 8 (Fig S6F and G). These results are likely because of the expression of different isoforms of Drp1 in these cell lines. Furthermore, we observed a slightly different enrichment of potential interaction partners of Drp1 (Tables S4 and S5).

In summary, these results showed that the selected Nbs recognize different domains of Drp1 and can be readily converted into functional capture reagents to precipitate endogenous Drp1 together with its interaction partners.

### bivD7 enables immunofluorescence detection of endogenous Drp1

To further examine the potential of the Drp1-Nbs as research tools, we investigated their performance in immunofluorescence (IF) microscopy. First, we applied D7 and D63 as primary binding molecules in combination with a fluorescently labelled anti-VHH antibody in fixed and permeabilized U2OS cells transiently over-expressing GFP-Drp1. Only D7 showed colocalization with GFP-Drp1,

whereas D63 showed no staining (Fig S7), suggesting that this Nb does not bind Drp1 in IF. Furthermore, when we tested IF for endogenous Drp1 in U2OS cells, we did not observe specific Drp1 staining for either Nb (Fig 4A). Considering that a bivalent format can be superior to monovalent versions for imaging purposes (Virant et al, 2018; Fagbadebo et al, 2022), we genetically fused two D7 or two D63 Nbs head-to-tail connected by a flexible Gly-Ser linker [(G₄S)₄] and generated a bivalent format of D7 (bivD7) or D63 (bivD63), respectively. In addition, we inserted a sortase-tag for site-directed labelling. Both bivalent Nbs were purified as secreted proteins from ExpiCHO cells (Fig S8A) followed by measuring their binding affinities as described for the monovalent versions. The results showed that bivD7 had an increased affinity in the sub-nanomolar range due to decelerated dissociation, whereas the affinity of bivD63 did not increase compared with the monovalent version (Fig S8B and C). Initially, we tested both bivalent Nbs for colocalization with GFP-Drp1 in U2OS cells and observed more specific signals compared with the monovalent versions, with bivD7 tending to recognize Drp1 more potently compared with bivD63 (Fig S9). Next, we applied bivD7 and bivD63 for staining of endogenous Drp1. Notably, only the bivD7 showed a clear colocalization with the antibody-labelled Drp1 in U2OS cells (Fig 4B). To further confirm that bivD7 specifically recognizes Drp1, we performed immunofluorescence (IF) imaging of wt and Drp1 KO HeLa cells. Whereas a colocalization with the Drp1 antibody signal was detected in wt HeLa cells, no Nb-derived signal was observed in Drp1 KO HeLa cells (Figs 4C and S10A and B). From these results, we concluded that the bivalent format of D7 is a suitable probe for IF applications to visualize endogenous Drp1.

## bivD7 is suitable for SRM

Because of their small size and their ability to access dense cell compartments and structures, Nbs have been previously described as highly versatile probes for SRM (Koch & Brocard, 2012; Virant et al, 2018; Cramer et al, 2019; Götzke et al, 2019; Maidorn et al, 2019; Driouchi et al, 2022). Hence, we tested the bivD7 in SRM using stochastic optical reconstruction microscopy (STORM). For site-directed labelling, we introduced a peptide comprising an azide group at the C-terminus by chemoenzymatic sortagging (Popp & Ploegh, 2011; Virant et al, 2018), followed by addition of a dibenzocyclooctyne (DBCO) derivative by click chemistry (Fagbadebo et al, 2022) which allowed us to flexibly and specifically conjugate STORM-compatible dyes such as AlexaFluor 647 (AF647). For initial validation, we stained wt HeLa and Drp1 KO HeLa cells with the fluorescently labelled bivD7$_{AF647}$, confirming its functionality as an imaging probe (Fig S11). In addition, we stained Drp1 in a cell line expressing monomeric enhanced GFP (mEGFP)-tagged Drp1 (mEGFP-Drp1) at endogenous expression levels. Confocal imaging and colocalization analysis revealed a reasonable overlap of the mEGFP-Drp1 and the bivD7$_{AF647}$ signal with a Pearson's correlation coefficient of 0.73 ± 0.07 (Fig 5A–C). Next, we imaged Drp1 stained with bivD7$_{AF647}$ in mEGFP-Drp1 U2OS cells using STORM (Fig 5D). The reconstituted bivD7$_{AF647}$ signal overlaid with the epifluorescence signal of mEGFP-Drp1 acquired in the same cell before STORM imaging as expected. These results demonstrate the applicability of the bivD7$_{AF647}$ as an imaging probe with enough brightness and specificity to label Drp1 at endogenous expression levels. Finally, we imaged bivD7$_{AF647}$-stained Drp1 in wt U2OS cells by STORM. Here, we detected localizations of both diffuse cytosolic Drp1 as well as mitochondrial Drp1 complexes (Fig 5E and F). Utilizing the bivD7$_{AF647}$ for STORM imaging allowed us to resolve macromolecular assemblies of Drp1 at mitochondria demonstrating its usability as a specific Drp1 probe for SRM imaging with minimal linkage error (Fig 5G).

## Drp1-Cbs visualize the dynamics of endogenous Drp1 in live cells

To further visualize endogenous Drp1 in a "tag-free" approach in living cells, we generated Cb expression constructs by genetic fusion of D7 and D63 with TagRFP via a flexible Gly-Ser linker (Fig 6A). First, we tested the intracellular binding properties of the newly generated Drp1-Cbs by intracellular immunoprecipitation (IC-IP) (Maier et al, 2015; Traenkle et al, 2015). To this end, we transiently transfected HEK293 cells with the Drp1-Cbs or an unrelated Cb (Pep Cb, [Traenkle et al, 2020]) as a negative control (NC), precipitated the Cbs with a RFP-Trap, and analysed the bound fraction for endogenous Drp1. Immunoblot analysis showed enrichment of endogenous Drp1 along with the precipitated Drp1-Cbs, but not with the negative control (Fig 6A), from which we concluded that both Drp1-Cbs retain their binding properties upon intracellular expression. When using Cbs as intracellular probes in general, it is important to be aware that the fluorescent emission signal originating from the unbound Cbs may mask the signals of the antigen-bound Cbs. This is particularly a problem for the detection of Drp1, as Drp1 is predominantly localized in the cytosol and

unbound cytosolic Cbs result in a high background. This limits the specific detection of Drp1 and, for example, its recruitment to mitochondrial fission sites. To improve the signal-to-noise ratio, we further modified our Drp1-Cbs and generated turnover-accelerated versions that are subject to faster proteasomal degradation when unbound, as previously described (Keller et al, 2018). We tested the applicability of this approach by expressing both standard and turnover-accelerated D7 and D63 Cbs in wt HeLa and Drp1 KO HeLa cells. Quantitative imaging of living cells using nuclear-localized GFP for transfection control and cell segmentation revealed significantly lower average cellular intensities and thus a lower cytosolic background originating from the unbound Cb fraction in the case of turnover-accelerated Cbs compared with their unmodified counterpart (Fig 6B and C). Moreover, in Drp1 KO HeLa cells the signal of turnover-accelerated Cbs was further reduced because of the lack of antigen (Fig 6B and C). This decrease indicates that the modification significantly reduces the amount of unbound Drp1-Cbs and increases the signal-to-noise ratio of these intracellular imaging probes. Finally, we analysed the potency of turnover-accelerated Drp1-Cbs to visualize the dynamic relocalization of Drp1 from the cytosol to the mitochondria after chemical induction of mitochondrial fission. To induce fission and thus, recruitment of Drp1 to mitochondria, we treated U2OS cells transiently expressing unmodified or turnover-accelerated Drp1-Cbs with carbonyl cyanide m-chlorophenylhydrazone (CCCP), an uncoupler of mitochondrial oxidative phosphorylation. CCCP was previously reported to induce fission and Drp1 translocation to mitochondria (Voccoli & Colombaioni, 2009; Jones et al, 2017; Park et al, 2018; Pascucci et al, 2021). Time-lapse imaging of cells expressing the unmodified Drp1-Cbs showed rather diffuse Cb signals that did not change over time. In contrast, in cells expressing the turnover-accelerated Drp1-Cbs, an increasing number of spots became visible within the observation period (Fig 7), which was not observable in the absence of CCCP (Fig S12). Based on the colocalization with the MitoTracker signal, these characteristic spots strongly suggest an accumulation of Drp1 at mitochondrial fission sites. Notably, we did not observe such patterns in Drp1 KO HeLa cells, which further underlines that the detected spots are most likely Drp1 complexes bound by the Drp1-Cbs (Fig S13). In summary, these findings indicated that turnover-accelerated Drp1-specific Cbs are suitable for monitoring and visualizing the dynamic localization of endogenous Drp1 at the MOM after induction of mitochondrial fission in living cells.

# Discussion

There is increasing evidence of dysregulated Drp1 in the context of a variety of diseases, including neurodegeneration such as Parkinson's (Han et al, 2020), cardiovascular diseases (Zunino et al, 2007; Sharp et al, 2014; Kim et al, 2015) and cancer (Rehman et al, 2012; Kashatus et al, 2015; Serasinghe et al, 2015). However, the exact molecular mechanisms of Drp1 function and its dynamic regulation remain to be elucidated. Therefore, the development of advanced research tools to study this key player in

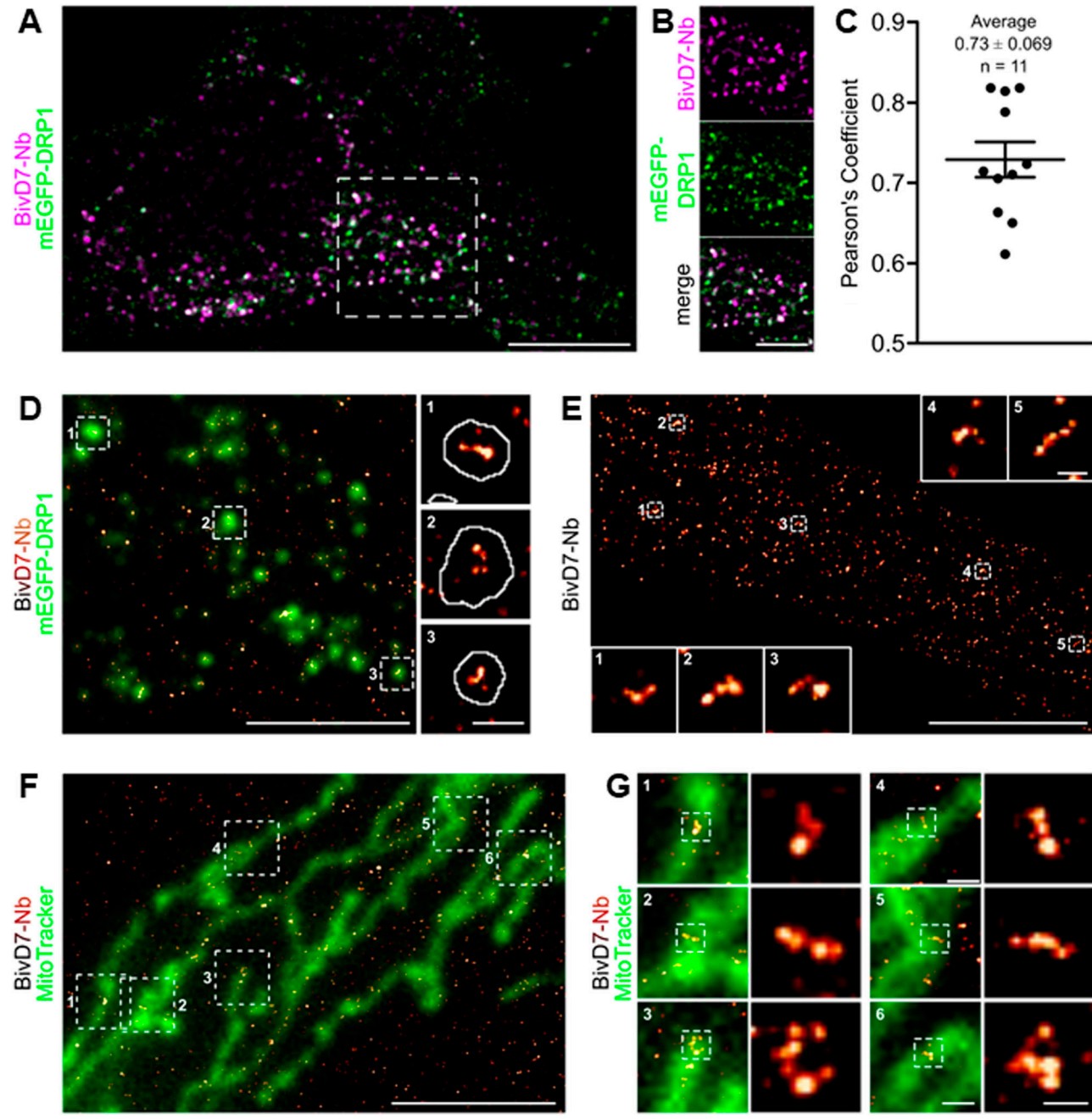

**Figure 5. Super-resolution imaging of Drp1 in U2OS cells.**
**(A)** Representative confocal fluorescence microscopy image of U2OS mEGFP-Drp1 cells (mEGFP-Drp1 signal shown in green) stained with bivD7 directly conjugated to AlexaFluor647 (bivD7$_{AF647}$; magenta). **(B)** Zoomed images correspond to crop regions as indicated in (A). Scale bar 10 $\mu$m, crop 5 $\mu$m. **(C)** Pearson's correlation coefficient of mEGFP and bivD7$_{AF647}$ fluorescence emission signals calculated from background-corrected images shown in (A). Data are representative for n = 3 independent experiments with n = 11 cells. **(D)** Reconstructed super-resolution image of Drp1 stained with bivD7$_{AF647}$ (orange-hot) overlayed with the epifluorescence signal of mEGFP-Drp1 (green). Zoomed images (right) correspond to cropped regions as indicated. White line in the zoomed regions corresponds to the outline of the thresholded mEGFP-Drp1 signal. Images are representative for n = 3 independent experiments. Scale bar 5 $\mu$m, zoom 500 nm. **(E)** Reconstructed super-resolution image of Drp1 labelled with bivD7$_{AF647}$ in wt U2OS cells. Insets correspond to zoomed regions as indicated. Scale bar 5 $\mu$m, insets 200 nm. **(F)** Reconstructed super-resolution image of Drp1 labelled with bivD7$_{AF647}$ (orange-hot) overlayed with an epifluorescence microscopy image of mitochondria stained with MitoTracker (green) in wt U2OS cells. Scale bar 5 $\mu$m. **(G)** Zoomed areas of the image shown in (F) (left images) with further zoom on Drp1 assemblies (right images) as indicated. Scale bar 500 nm, zoomed images 200 nm. **(D, E, F, G)** Data are representative of at least n = 3 independent experiments.

mitochondrial fission is urgently needed (Giacomello et al, 2020; Montecinos-Franjola et al, 2020; Tong et al, 2020; Zerihun et al, 2023).

Here, we present, to the best of our knowledge for the first time, an Nb-based toolset for studying endogenous Drp1 in different biochemical and cell biological research settings. We identified two

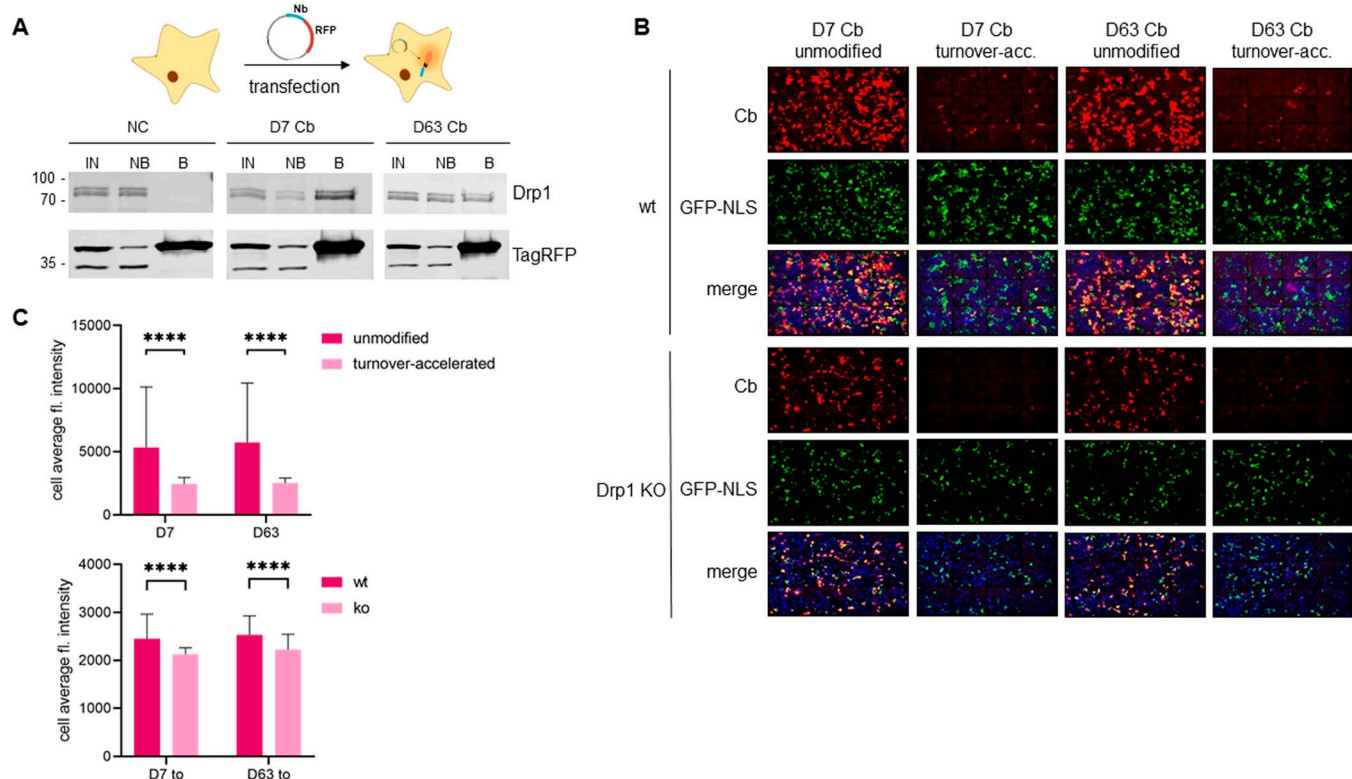

**Figure 6. D7 and D63 Cb recognize their antigen and are stabilized in the presence of Drp1 in living cells.**
**(A)** Intracellular immunoprecipitation of Drp1. Soluble protein fractions of HEK293 cells expressing the indicated chromobodies (D7 Cb, D63 Cb or as negative control an unrelated chromobody [NC]) were subjected to immunoprecipitation with the RFP-Trap. Input (IN), non-bound (NB) and bound fractions (B) were analysed by immunoblotting with an anti-Drp1 antibody (upper panel) and an anti-TagRFP antibody (lower panel). Molecular weights in kD on the left. **(B)** Representative fluorescence images of living wt HeLa or Drp1 KO HeLa cells transiently expressing either the unmodified or the turnover-accelerated Drp1-Cb constructs. As transfection control, a GFP-NLS encoding construct was co-transfected. Nuclei were stained with Hoechst33258. **(C)** Bar chart representing the cell average fluorescence intensity of Cb signal (red channel) quantified from (B). For the comparison of unmodified and turnover-accelerated Cbs, significance was calculated using Kruskal Wallis test with Dunn's multiple comparisons test (upper panel). For the comparison of the turnover-accelerated Cb (D7 to; D63 to) intensities in wt HeLa cells and Drp1 KO cells, the significance was calculated using two-way ANOVA with Sidak's multiple comparisons (lower panel). Data are presented as mean ± SD; n. s. $P > 0.05$, ****$P < 0.0001$; n > 130 cells each.

stable and well-producible Nbs, D7 and D63, which bind distinct domains of Drp1 with high affinity. Notably, binding of D63 to the C-terminal GED of Drp1 induces an increased GTPase activity in vitro. It can be speculated that this is because of a stabilization of the intramolecular interaction between the GED and the N-terminal GTP-binding domain—a conformation known to be important for the enzymatic function of Drp1 (Zhu et al, 2004; Frohlich et al, 2013). By targeted functionalization and engineering, we have successfully applied the identified Nbs as (i) affinity capture tools (Drp1 nanotraps), (ii) labelling probes for fluorescence microscopy including SRM, and (iii) Cbs for visualization of Drp1 in living cells. For affinity capture from whole cell lysates, we demonstrated that both Drp1 nanotraps efficiently precipitated Drp1. Notably, immunoblot analysis with a monoclonal Drp1 antibody showed captured Drp1 in the form of several bands. These findings are comparable with previous reports also showing multiple bands for Drp1 (Loson et al, 2013; Yu et al, 2019; Xie et al, 2020; Yu et al, 2021), which can be attributed to the size differences between individual Drp1 isoforms (Chen et al, 2000; Itoh et al, 2018). Applying the nanotraps to precipitate Drp1 from other cell types or even tissues, might result

in the identification of a different Drp1 pattern reflecting the expression of different isoforms. Indeed, we observed an enrichment of different Drp1 isoforms when we used the Drp1 nanotraps for immunoprecipitation studies in different cell lines. With in-depth MS analysis of the proteome of both Drp1 nanotraps derived from the soluble fraction of three different cell lines, we confirmed the effective and specific enrichment of Drp1 with a high sequence coverage based on the identified peptides. It is noteworthy that the MS analysis also revealed the enrichment of MOM- and peroxisome-associated factors. Interestingly, the latter were found exclusively for the D63 nanotrap, but not for the D7 nanotrap, which could be explained by steric and allosteric effects caused by the binding of the Nbs at the different domains of Drp1. Overall, both Drp1 nanotraps precipitated endogenous Drp1 more efficiently than the conventional monoclonal Drp1 antibody. Thus, we conceive that both Nb-based affinity matrices enable a fast and efficient investigation of the Drp1-interatome under different physiological conditions and/or in different tissues.

Despite their high affinity for recombinant Drp1, both Nbs initially failed to stain endogenous Drp1 by immunofluorescence labelling.

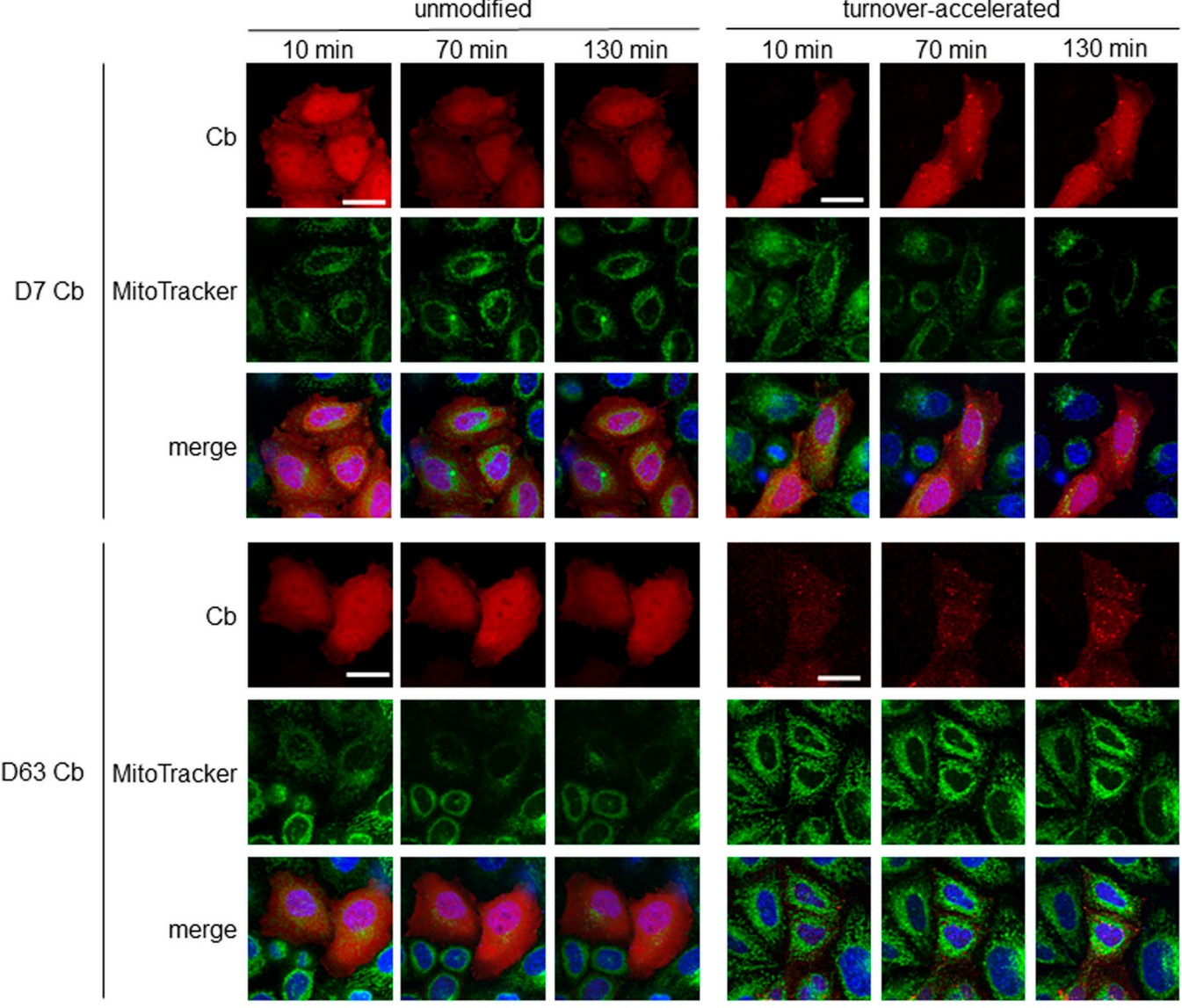

**Figure 7. Turnover-accelerated Cbs monitor relocalization of Drp1 in living cells.**
Fluorescent time-lapse microscopy of wt HeLa cells transiently expressing unmodified or turnover-accelerated chromobodies (Cbs) (red). 24 h post transfection, cells were stained with Hoechst33258 (blue) and MitoTracker green (green) and subsequently treated with 20 μM cyanide m-chlorophenylhydrazone (CCCP). Representative confocal images were taken after 10, 70, and 130 min of treatment. For each Cb, brightness and contrast were individually set and kept for all time points. Scale bar 25 μm.

This might be because a lower accessibility of the epitope or a higher off-rate of the Nbs for Drp1 in its native environment. To improve their binding properties by increasing avidity, we have generated bivalent Nbs, a format that has previously significantly improved antigen recognition for other Nbs (Virant et al, 2018; Fagbadebo et al, 2022). Accordingly, we found that bivD7 is able to specifically stain endogenous Drp1 in fixed cells. Most importantly, by site-directed fluorophore conjugation, we generated a probe with a substantially smaller linkage error compared with conventional antibodies, applicable for super-resolution STORM imaging of Drp1. Commonly applied labelling strategies for SRM include direct (labelled primary antibody) and indirect (labelled secondary antibody) immunofluorescence techniques or genetically encoded fluorescent protein fusions. The linkage error, which describes the distance between the biological sample and the detectable moiety, is determined by the size of the structure used to label the protein of interest (e.g., the antibody) and the random distribution of fluorescence emitters on this structure. This can induce a fluorescence displacement of up to ~30 nm causing mislocalization of the target structure coordinates (Carrington et al, 2019; Fruh et al, 2021). As demonstrated, the bivD7 conjugated to AlexaFluor647 enables for the first time super-resolved imaging of endogenous Drp1 complexes located in the cytoplasm and on mitochondria.

Recently, it was reported that labelling of Drp1 with genetically fused tags including fluorescent proteins such as GFP alters its

oligomerization state and consequently affects its function (Montecinos-Franjola et al, 2020). Therefore, a tag-free approach is desirable to minimize artefacts when studying Drp1 in living cells. In this context, transiently binding Cbs are known as versatile probes to visualize endogenous antigens in living cells with high spatial and temporal resolution (Rothbauer et al, 2006; Burgess et al, 2012; Maier et al, 2015; Panza et al, 2015; Wegner et al, 2017). Here, we reformatted D7 and D63 into intracellularly functional Cbs and monitored the dynamics of endogenous Drp1 by fluorescent time-lapse imaging of living cells. Considering that Drp1 is mainly diffusely localized in the cytoplasm, we had to face the challenge of distinguishing between the signals originating from Drp1-binding Cbs and unbound Cbs, a problem that is constantly arising when using genetically encoded imaging probes expressed from a constitutive promoter (Wagner & Rothbauer, 2020). Several approaches have been developed to reduce the levels of unbound intrabodies and Cbs (Sibler et al, 2005; Gross et al, 2013; Tang et al, 2016; Keller et al, 2018). Here, we introduced an N-terminal arginine residue, which was previously reported to convey a fast proteasomal degradation of unbound Cbs in the cytoplasm via the N-end rule (Varshavsky, 2005; Keller et al, 2018). The improved signal-to-noise ratio of these turnover-accelerated Drp1-Cbs now enables for the first time the visualization of the drug-induced relocalization of endogenous cytoplasmic Drp1 towards mitochondrial fission sites in living cells. In addition to imaging, the Nbs can be further adapted as building blocks of, for example, Drp1 modulating functional intrabodies for extended research applications. As an example, we started to develop Drp1-Nbs as inducible degrons for targeted degradation of Drp1 in living cells. In summary, currently shown as a proof-of-principle study, we are convinced that the herein-presented Nbs open up new opportunities for comprehensive and detailed studies of Drp1 and its pathophysiological role as a key regulator of mitochondrial fission in advanced experimental settings.

# Materials and Methods

### Nanobody library construction

Alpaca immunization with recombinant Drp1 as well as Nb library generation was performed as described before (Rothbauer et al, 2006). Animal immunization was authorized by the government of Upper Bavaria (Permit number 55.2-1-54-2532.0-80-14). Briefly, an alpaca (*V. pacos*) was immunized with recombinant human Drp1 (Drp1). An initial priming dose of 0.48 mg Drp1 was followed by booster injections of 0.24 mg each in the 3rd, 4th, 7th, and 12th wk. Serum samples of 20 ml were taken in the 9th wk of immunization for analysis of seroconversion by ELISA. 13 wk after the first dose, 100 ml blood was drawn, and the lymphocytes isolated by Ficoll gradient centrifugation using the Lymphocyte Separation Medium (PAA Laboratories GmbH). Total RNA was isolated using TRIzol (Life Technologies) followed by reverse transcription of mRNA into cDNA using a First-Strand cDNA Synthesis Kit (GE Healthcare). To isolate the Nb repertoire, three nested PCR reactions were performed with the following primers: (i) CALL001 and CALL002, (ii) forward primer

set FR1-1, FR1-2, FR1-3, FR1-4 and reverse primer CALL002, and (iii) forward primer FR1-ext1 and FR1-ext2 and reverse primer set FR4-1, FR4-2, FR4-3, FR4-4, FR4-5, and FR4-6 to introduce SfiI and NotI restriction sites. The sequences for all primers (synthesized by Integrated DNA Technologies) used in this study are listed in Table S6. The amplified comprising the Nb repertoire was then subcloned into the pHEN4 phagemid vector (Arbabi Ghahroudi et al, 1997) using the SfiI/NotI restriction sites thereby generating the Drp1-Nb library.

### Nanobody screening

To select for Drp1-specific Nbs, *E. coli*, TG1 cells containing the Drp1-Nb library in pHEN4 were infected with the M13K07 helper phage. Next, the hereby generated Nb-presenting phages were isolated from culture supernatant by PEG precipitation and $1 \times 10^{11}$ phages were used for subsequent biopanning. In each selection round, extensive blocking was carried out with 5% milk or BSA in PBST (PBS, 0.05% Tween 20, pH 7.4) (Pardon et al, 2014). To deplete non-specific binders, phages were first added to Nunc Immuno MaxiSorp tubes (Thermo Fisher Scientific) coated with 10 µg/ml GFP before transferring them to tubes coated with 10 µg/ml Drp1 or 10 µg/ml GFP as non-related antigen for 2 h at RT. Next, unbound phages were removed by washing with stringency increasing each round. Bound phages were eluted with 100 mM triethylamine (pH 12.0) and immediately neutralized with 1 M Tris–HCl pH 7.4. After each round of panning, log phase *E. coli* TG1 cells were infected with the eluted phages and grown on selection plates to rescue enriched phages. Drp1-specific phage enrichment was monitored by determining the CFUs after each round. After two rounds of panning, 260 individual clones were randomly selected and screened by phage ELISA using immobilized Drp1 or GFP as control. HRP-labelled anti-M13 monoclonal antibody (GE Healthcare) and 1-Step Ultra TMB solution (Thermo Fisher Scientific) were used for detection of bound phages. The reaction was stopped with 100 µl of 1 M $H_2SO_4$ and the signal was detected on a plate reader (Pherastar) at 450 nm. Phage ELISA-positive clones were defined by a twofold signal above the control.

### Expression plasmids

For bacterial Nb expression, Nb encoding cDNA were cloned into the pHEN6C vector as previously described (Rothbauer et al, 2008). For mammalian expression of bivalent Nbs, cDNAs comprising the two coding sequences of the respective Nbs connected by a flexible Gly-Ser linker $[(G_4S)_4]$ and a C-terminal sortase tag (L-P-E-T-G) followed by a His6-tag were generated and inserted into pCDNA3.4 expression vector variant comprising N-terminal signal peptide for secretion (Wagner et al, 2021) as following: For bivD7, each Nb sequence was amplified separately by PCR using bivD7-1 and bivD7-2 or bivD7-3 and downEcoRI-rev primer pairs. Next, both Nbs were fused by overlap-extension PCR using the primers bivNshort_for and downEcoRI-rev. For bivD63 cloning, the primer pairs bivD63forN and bivNtermGS-rev as well as bivD63forC and downEcoRI-rev were used followed by bivD63_fusion and downEcoRI-rev for overlap-extension PCR. Finally, the resulting sequence was inserted into pCDNA3.4 via restriction digestion with

Esp3I and EcoRI. To generate unmodified Cbs, Nb sequences were cloned into pTagRFP by BglII and HindIII restriction digest as previously described (Panza et al, 2015). For turnover-accelerated Cbs, the Nb sequence was inserted via PstI and BspEI into the previously described pEGFP-Ubi-R-3xGS-tagRFP plasmid (Keller et al, 2018). The coding sequence of Drp1 was amplified from pAcGFP-Drp1 plasmid (Jenner et al, 2022) with Drp1_GTPaseFor and Drp1_C_term_rev and cloned into pEGFP-N1 backbone. For deletion constructs for domain mapping, all sequences were amplified from AcGFP-Drp1 and cloned into pEGFP-N1 with Drp1-GTPaseFor forward primer and the respective reverse primer: Drp1-VD-rev (GTPase-MD-VD), Drp1-MD-rev (GTPase-MD), Drp1-GTPase-rev (GTPase). The correct sequences of all constructs were confirmed by Sanger sequencing. All expression constructs used in this study are listed in Table S7.

## Protein expression and purification

Monovalent Drp1-Nbs were expressed and purified as previously described (Rothbauer et al, 2008; Wagner et al, 2021). Bivalent Drp1-Nbs were produced using the ExpiCHO system (Thermo Fisher Scientific) according to the manufacturer's instructions (Wagner et al, 2021; Fagbadebo et al, 2022). Purity of produced Nbs was assessed using SDS–PAGE. Thus, protein samples were denatured (10 min, 95°C) with 2x SDS-sample buffer (60 mM Tris–HCl, pH 6.8; 2% [wt/vol] SDS; 5% [vol/vol] 2-mercaptoethanol; 10% [vol/vol] glycerol, 0.02% bromphenol blue) and gels were stained with InstantBlue Coomassie (Abcam). Protein concentrations were determined using Bradford and spectrophotometer measurements. Drp1 was purified as described in (Jenner et al, 2022). Briefly, DRP1 (isoform 1) was expressed from a pTYB2 vector in *E. coli* BL21-CodonPlus (DE3)-RIPL. Bacteria were grown at 37°C in LB medium containing 100 $\mu$g/ml ampicillin and 35 $\mu$g/ml chloramphenicol to an $OD_{600\ nm}$ of 0.6. Protein expression was induced using 1 mM isopropyl 1-thio-$\beta$-d-galactopyranoside (IPTG) at 14°C for 18 h. Cell pellets were resuspended in 20 mM HEPES/KOH pH 7.4, 500 mM NaCl and 1 mM $MgCl_2$ containing 1 mM PMSF, 1 $\mu$g/ml DNAse I (Roche Diagnostics), and protease inhibitor (Complete; EDTA-free Protease Inhibitor Mixture; Roche Applied Science), homogenized by mechanical rupture and cell debris was removed by centrifugation. The supernatant was purified using chitin resin affinity purification (New England Biolabs, Inc.). Drp1 was eluted by cleavage from the affinity resin using 30 mM DTT at pH 8 for 48 h at 4°C and further purified using anion exchange chromatography (HiTrap Q FF, GE Healthcare) at pH 8. Pure elution fractions were pooled and dialyzed against 20 mM HEPES/KOH pH 7.4, 500 mM NaCl and 1 mM $MgCl_2$. Purity was assessed using SDS–PAGE as described and protein concentration was determined using Bradford and spectrophotometer measurements. Purified protein was stored with 20% (vol/vol) glycerol.

## Affinity measurements by biolayer interferometry

Binding kinetics analysis of Drp1-Nbs was performed using the Octet RED96e system (Sartorius) according to the manufacturer's recommendations. In brief, streptavidin-coated biosensor tips (SA,

Sartorius) were incubated for 40 s with 1.7–10 $\mu$g/ml biotinylated Drp1-Nbs diluted in Octet buffer (HEPES, 0.1% BSA). In the association step, a four-step twofold dilution series of Drp1 starting from 125 nM (D7), 62.5 nM (D63) or 120 nM (bivD7 and bivD63) were applied for 300 s followed by dissociation in Octet buffer for 720 s. Each concentration was normalized to a reference applying Octet buffer only for association. Data were analysed with the Octet Data Analysis HT 12.0 software using the 1:1 ligand model and global fitting.

## Drp1 GTPase activity measurements

GTPase activities for Drp1 were measured using a continuous, regenerative assay described by (Ingerman & Nunnari, 2005). In brief, Drp1 (100 nM final) was incubated with 0, 10, 100 or 1,000 nM D7, D63, or GFP Nb as negative control. Upon addition of 1 mM DTT, 1 mM PEPK, 600 $\mu$M NADH, ≥20 U/ml PK/LDH (all Sigma-Aldrich) and 1 mM GTP, GDP (both Jena Bioscience), or no nucleotide, the reactions were started with the final buffer containing 20 mM Hepes pH 7.4, 150 mM KCl, 2 mM $MgCl_2$. Upon thermal equilibration of the microtiter plates, NADH absorbances were measured for 120 min using a Safire Tecan-F129013 microplate reader (Tecan) operating at 340 nm and 37°C. Using BSA without nucleotide or GDP for rapid NADH depletion, absorbances were converted into NADH concentrations. GTPase rates were calculated using linear fits between 30 and 120 min of the traces.

## Cell culture and transfections

U2OS and HEK293 cell lines were obtained from ATCC (CRL3216, HTB-96); the HeLa Kyoto cell line (Cellosaurus no. CVCL_1922) was acquired from S. Narumiya (Kyoto University, Japan). Drp1 KO HeLa cells were generated in and provided by the laboratory of Thomas Langer (Max Plack Institute, Cologne, Germany). For mycoplasma testing, the mycoplasma kit Venor GeM Classic (Minerva Biolabs) and Taq polymerase (Minerva Biolabs) were applied. No additional authentication was performed as this study does not contain any cell-line-specific experiments. Culturing of cell lines was carried out by standard protocols. In brief, cells were grown in DMEM high glucose, pyruvate, GlutaMax (Thermo Fisher Scientific) with 10% (vol/vol) FCS (Thermo Fisher Scientific) and 1% (vol/vol) penicillin/streptomycin (Thermo Fisher Scientific). For routine passaging of cells, 0.05% trypsin–EDTA (Thermo Fisher Scientific) was used. Cells were cultivated at 37°C and 5% $CO_2$ in a humidified incubator. U2OS cells were transiently transfected with Lipotectamine 2000 (Thermo Fisher Scientific) according to the manufacturer's recommendations. HEK293 and HeLa cells were transfected with Polyethyleneimine (PEI, Sigma-Aldrich) as previously described (Braun et al, 2016).

## Nanobody immobilization on NHS-sepharose matrix

1.2 mg Drp1-Nbs per 1 ml NHS-Sepharose (Cytiva) were immobilized according to the manufacturer's instructions.

## Sortase labelling of nanobodies

For sortase coupling, 50 μM Nb, 250 μM sortase peptide (H-Gly-Gly-Gly-propyl-azide, synthesized by AG Maurer, University of Tübingen) dissolved in sortase buffer (50 mM Tris, pH 7.5, and 150 mM NaCl) and 10 μM sortase were mixed in coupling buffer (sortase buffer with 10 mM CaCl$_2$) and incubated for 4 h at 4°C. Uncoupled Nb and Sortase were removed by immobilized metal affinity chromatography, and the excess peptide was depleted via Amicon Ultra Centrifugal Filters (Merck Millipore). To label the azide-coupled nanobody with a fluorophore, SPACC (strain-promoted azide–alkyne cycloaddition) click chemistry reaction was applied. For this, fivefold molar excess of dibenzocyclooctyne-AlexaFluor647 (DBCO-AF647) (Jena Bioscience) was added to the azide-coupled Nb followed by incubation for 2 h at room temperature and subsequently purified by dialysis and hydrophobic interaction chromatography.

## Mammalian cell lysis and protein extraction

For intracellular immunoprecipitation (IC-IP), 2.5–3 × 10$^6$ HEK293 cells were seeded in a 100 mm culture dish (P100). The next day, cells were transfected with the chromobody-encoding plasmid. After 24 h, cells were harvested at 90% confluency. This workflow was also followed for the domain mapping constructs. For immunoprecipitations (IPs) using nanotraps, T175 flasks or P100 dishes of HEK293, U2OS or HeLa cells were harvested at ~90% confluency. HEK293 cells were detached by light pipetting and U2OS or HeLa cells by trypsinization. Subsequently, the cells were washed with 1 ml PBS, and the pellets were stored at –80°C until lysis. For lysis, T175 flask pellets were resuspended in 600 μl, P100 pellets in 200 μl lysis buffer (50 mM Tris–HCl, pH7.5; 150 mM NaCl; 1 mM EDTA, pH 8; 0.5% TritonX-100; 1 μg/μl DNaseI; 2.5 mM MgCl$_2$; 2 mM PMSF; 1x Protease inhibitor Mix M [Serva]). The lysate was passed through 20G, 23G, and 27G needle gauges multiple times with intermediate vortexing and 10 min incubation steps on ice. Subsequently, the lysate was centrifuged for 10 min at 18,000$g$ at 4°C. The supernatant was transferred to a fresh tube and diluted 1.5-fold with dilution buffer (50 mM Tris–HCl pH 7.5; 150 mM NaCl; 1 mM EDTA, pH 8; 2 mM PMSF).

## Immunoprecipitation

Soluble protein extracts were generated from HEK293, HeLa and U2OS cells as described above. For analysis of the input (IN), 20 μl of the lysate was mixed with an equal amount of 4x SDS-Sample buffer. For IP, 80 μl slurry of each nanotrap was equilibrated in IP buffer (50 mM Tris–HCl pH 7.5; 150 mM NaCl; 1 mM EDTA, pH 8) and added to equal amounts of lysate. As positive (domain mapping) or negative control (precipitation of endogenous Drp1), 80 μl GFP-Trap slurry (ChromoTek) was used. For IC-IP, 80 μl of RFP-Trap slurry (ChromoTek) was used. Soluble protein extracts were incubated with the nanotraps overnight at 4°C on an end-over end rotor. For positive control (precipitation of endogenous Drp1), 4 μl anti-Drp1 antibody was added to the lysate and after overnight incubation at 4°C 80 μl equilibrated Protein A/G sepharose slurry were added followed by additional 3–4 h incubation at 4°C. Beads were harvested by centrifugation (2,000$g$,

2 min, 4°C) and supernatant (non-bound sample) was mixed 1:1 with 4x SDS-sample buffer. The beads were washed three times with 500 μl wash buffer (IP buffer with 2 mM PMSF, for MS samples in addition to 1x protease inhibitor mix and 0.05% Tween20). With the third wash, beads were transferred into a fresh tube. The beads were boiled in 60 μl 2x SDS-sample buffer for 10 min at 95°C and the supernatant was transferred into a fresh tube (bound sample).

For immunoblotting of IP samples, 1% of input and non-bound and 33% of the bound sample were loaded on an SDS-Page and blotted on nitrocellulose membrane as previously described (Fagbadebo et al, 2022). Immunoblots were probed with the following antibodies: anti-Drp1, anti-TagRFP, anti-GAPDH (Fig S14A–C). All antibodies used in this study are listed in Table S8. For immunoblot scanning, a Typhoon-Trio laser scanner (GE Healthcare) was used with excitation 633 nm and emission filter 670 nm BP 30 settings.

## Chromobody imaging

wt HeLa cells (8,000 cells/well) or Drp1 KO HeLa cells (11,000 cells/well) were seeded in μclear 96 well plates (Greiner Bio-One). The next day, cells were transfected with the respective plasmids. 24 h post transfection, cells were stained with 400 nM MitoTracker green (Life Technologies) and 4 μg/ml Hoechst33258 for 30 min. Afterwards, medium was replaced with live-cell visualization medium DMEMgfp-2 (Evrogen) supplemented with 10% FBS, 2 mM L-glutamine with or without 20 μM CCCP. Images were taken 10, 70, and 130 min after compound addition with ImageExpress Micro Confocal High Content screening system (Molecular devices) at 40x magnification. During imaging, cells were kept at 37°C and 5% CO$_2$. Raw microscopy images were adjusted in brightness and contrast using MetaExpress software (64 bit, 6.2.3.733 or higher, Molecular devices).

## Immunofluorescence in fixed cells

Cells were seeded in μclear 96 well plates (Greiner Bio-One) in the same densities as described for Cb imaging. The next day, cells were transfected with GFP-Drp1 plasmid or left untransfected. 24 h post-transfection, cells were washed twice with PBS and subsequently fixed with 3.7% PFA for 10 min at RT. After three times washing with PBS, cells were incubated with TBST for 10 min at RT. After additional washes, cells were blocked with 5% BSA in TBST for 1 h. Afterwards, the cells were incubated with 100–1,000 nM Nb and anti-Drp1 antibody (1:100–1:300) in 5% BSA/TBST overnight at 4°C. Then, unbound Nb and antibody were washed away and the cells incubated with secondary antibody (anti-rabbit labelled with AF647; 1:1,000) and Cy5-conjugated goat anti-alpaca antibody (1:500) for 1 h at RT in 2.5% BSA/TBST. Nuclei staining with 4′, 6-diamidino-2-phenylindole (DAPI; Sigma-Aldrich) was performed concurrently. After additional washing with TBST and PBS, images were acquired using an ImageExpress Micro Confocal High Content screening system (Molecular devices) at 40x magnification. Raw microscopy images were adjusted in brightness and contrast using MetaExpress software (64 bit, 6.2.3.733 or higher, Molecular devices).

## Liquid chromatography-MS sample preparation and measurement

Immunoprecipitation of Drp1 using the D7 and D63 nanotraps were compared with the GFP-Trap (negative control) in three technical replicates. Immunoprecipitation was performed as described above and proteins were subsequently separated by SDS–PAGE (4–12% NuPAGE tris gel [Invitrogen]) for 7 min at 200 V and stained with Coomassie blue. The protein gel pieces were subjected to tryptic digestion as described previously (Shevchenko et al, 2006). Purified peptide samples were measured on a Q Exactive HF mass spectrometer (Thermo Fisher Scientific) online-coupled to an Easy-nLC 1200 UHPLC (Thermo Fisher Scientific). Peptides were separated using a 20-cm-long, 75–$\mu$m–inner diameter analytical HPLC column (ID PicoTip fused silica emitter; New Objective) packed in-house with ReproSil-Pur C18-AQ 1.9-$\mu$m silica beads (Dr Maisch GmbH) and eluted using a 60 min segmented gradient from to 10–90% of solvents A (0.1% formic acid) and B (80% acetonitrile in 0.1% formic acid) at a constant flow rate of 200 nl/min. The column temperature was maintained at 40°C using an integrated column oven. Peptides were ionized by nanospray ionization and a capillary temperature of 275°C. The mass spectrometer was operated in the positive ion mode. Full MS scans were acquired in a range of 300–1,750 m/z at resolution of 60,000. Seven most intense multiple-charged ions were selected for HCD fragmentation with a dynamic exclusion period of 30 s and tandem MS (MS/MS) spectra were acquired at resolution of 15,000.

## Mass spectrometry data analysis

The acquired raw files were processed by MaxQuant software (version 2.0.3.0.) (Cox & Mann, 2008) and searched against the Uniprot *Homo sapiens* database (105.079 entries, downloaded 2022/08/01), *V. pacos* derived nanotraps and commonly observed protein contaminants. For MS and MS/MS, the peptide mass tolerance was set at 4.5 ppm and 20 ppm, respectively. Only two missed cleavages were allowed for the tryptic digestion. Carbamidomethylation (C) was used as fixed modification, whereas oxidation (M) and acetylation (protein N-term) were defined as variable modifications. False discovery rate was set to 1% at both peptide and protein levels. For label-free quantification, a minimum ratio count of two was requested. Intensity-based absolute quantification (iBAQ) was enabled. Downstream analysis of the "proteinGroups.txt" output table was performed in Perseus (version 1.6.15.0). Contaminants, reversed proteins and proteins only identified by site were filtered out. Only proteins that were quantified in two of three replicates of either the bait or control pulldown were retained in the dataset. Missing values were imputed (width 0.3, downshift 1.8). To determine significantly enriched proteins, a multi-volcano analysis (Hawaiian plot) was performed. The $s_0$ and FDR parameters of the multi-volcano analysis were: for class A (higher confidence, $s_0 = 0.1$, FDR = 0.01) and class B (lower confidence, $s_0 = 0.1$, FDR = 0.05). Additional graphical visualization was performed in the R environment (version 4.1.1) and in GraphPad (version 8.0.1.).

## Cell cultivation, seeding, and staining for super resolution microscopy

U2OS wt or U2OS mEGFP-Drp1 cells were cultivated in DMEM (low glucose) supplemented 10% (vol/vol) FBS and 1% (vol/vol) penicillin/streptomycin (Invitrogen) at 37°C and 5% (vol/vol) $CO_2$. For microscopy, the cells were seeded in 35 mm $\mu$-Dish 1.5H glass bottom imaging chambers (for confocal imaging) or on 35 mm 1.5H glass coverslips in in a six-well plate (for SMLM) at a density of 3 × $10^5$ cells per dish/well and grown for 18 h. If required, mitochondria were stained using 200 nM MitoTracker Green FM (Invitrogen) for 20 min. Cells were fixed in 4% (vol/vol) PFA in DMEM (pre-warmed to 37°C) for 10 min at RT and washed twice with PBS. To quench unreacted fixative, the cells were incubated in 50 mM $NH_4Cl$ for 15 min followed by permeabilization in 0.25% (vol/vol) Triton X-100 for 8 min. The cells were washed with PBS three times for 5 min and incubated with 1% (wt/vol) BSA in PBS for 1 h to block unspecific binding sites. DRP1 was labelled using 250 nM bivD7 coupled to AlexaFluor 647 (Invitrogen) in 1% (wt/vol) BSA in PBS over night at 4°C protected from light. The cells were washed three times with PBS and stored at 4°C in the dark until imaging.

## Confocal fluorescence microscopy

U2OS mEGFP-Drp1 cells were prepared as described above. Confocal fluorescence microscopy images were acquired on an inverted Infinity Line confocal dual-color STED 775 QUAD laser scanning microscope (Abberior Instruments) equipped with a UPLXAPO 60x/1.42 Oil objective (Olympus). mEGFP was excited at 488 nm and AF647 was excited at 640 nm wavelength. Images were acquired with a pixel size of 65 nm and a pixel dwell-time of 5 μs. The fluorescence emission signal was collected on avalanche photodiode detectors. Raw microscopy images were adjusted in brightness and contrast using Fiji/ImageJ (Schindelin et al, 2012). Colocalization of endogenous mEGFP-Drp1 and bivD7$_{AF647}$ was assessed by calculating the Pearson's correlation coefficient of the respective emission signals using Just Another Co-localization Plugin (JACoP) (Bolte & Cordelières, 2006) in Fiji/ImageJ (Schindelin et al, 2012). For this, the images were cropped to remove unspecific bivD7 signal outside of the cell and in the nucleus. The experiments are representative for n = 3 individual replicates with n = 11 cells in total.

## Single-molecule localization microscopy

U2OS WT cells were seeded and stained as described above and mounted on custom-made imaging chambers or single concave depression microscopy slides covered with 100–300 $\mu$l imaging buffer (50 mM Tris–HCl pH 8, 10 mM NaCl, 10% [wt/vol] glucose, 35 mM cysteamine [MEA], 0.5 mg/ml glucose oxidase [Sigma-Aldrich] and 40 $\mu$g/ml catalase [Sigma-Aldrich]) and sealed using Twinsil speed dental glue (Picodent). The samples were imaged on a home-built wide-field/TIRF microscope or on a SR GSD (3D) inverted DMI6000 B super-resolution wide-field/TIRF microscope (Leica Microsystems) equipped with a 100x, 1.47 N.A. GSD Objective, 405, 488, 532, and 642 nm excitation lasers and an Andor iXon Ultra 897 EMCCD camera. Image series were acquired with an exposure

time between 30 and 50 ms and an EM gain of 300. AF647 was excited using 640 nm wavelength and the 405 nm activation laser intensity was manually adjusted to keep a constant number of localizations per frame. Typically, 70,000–100,000 frames were recorded. Analysis was performed using the LAS X Software (Leica) and ThunderStorm (Ovesny et al, 2014). Images were rendered using a Gaussian with a width according to the localization precision. Image analysis was performed using Fiji/ImageJ (Schindelin et al, 2012). Images are representative for n = 6 individual experiments.

### Image analysis and statistics

Image analysis was performed with MetaExpress software (64 bit, 6.2.3.733 or higher, Molecular devices) for n > 130 cells per construct. Using the Custom editor (version 2.5.13.3 or higher) or the MetaExpress software, we developed an algorithm to determine the average signal intensity per transfected cell. Based on the assumption that cells usually are transfected with both constructs, the GFP-NLS signal was used as a marker for transfection and cell area. Cell average intensity of the TagRFP signal was determined for all GFP-positive cells. Using the DAPI signal in GFP-NLS negative cells, the TagRFP signal intensity was measured for background determination. The average background signal was determined (n = 2125 cells) and subtracted from all measurements. A workflow is shown in Fig S15A and B.

Mean and SD were calculated for each condition. For comparison of unmodified and turnover-accelerated Drp1-Cbs, significance was calculated using Kruskal Wallis test with Dunn's multiple comparisons test. For comparison of turnover-accelerated Cbs in wt HeLa and Drp1 KO HeLa cells, significance was calculated using two-way ANOVA with Sidak's multiple comparisons test. Tests were performed using GraphPad Prism software (Version 8 or higher). All graphs were prepared using GraphPad Prism.

## Data Availability

The data that support the findings of this study are available from the corresponding author upon reasonable request. The mass spectrometry proteomics data have been deposited to the ProteomeXchange Consortium via the PRIDE (Perez-Riverol et al, 2022) partner repository with the dataset identifier PXD051737.

## Supplementary Information

## Acknowledgements

This research was supported by the German Research Foundation (DFG) through RTG 2364 "MOMbrane" to T Froehlich, A Jenner; AJ Garcia-Saez, C Cavarischia-Rega, FO Fagbadebo,Y Lurz, E Schäffer, B Macek, and U Rothbauer. We thank the CECAD imaging facility for excellent assistance. We thank Prof. Steffen Jung (Weizmann Institute of Science) for his ongoing support and helpful discussions. We acknowledge support by Open Access Publishing Fund of University of Tuebingen.

## Author Contributions

T Froehlich: conceptualization, validation, investigation, visualization, and writing—original draft, review, and editing.

A Jenner: investigation, visualization, methodology, and writing—review and editing.

C Cavarischia-Rega: data curation, formal analysis, and writing—original draft, review, and editing.

FO Fagbadebo: investigation, methodology, and writing—review and editing.

Y Lurz: investigation, methodology, and writing—original draft, review, and editing.

DI Frecot: investigation, methodology, and writing—review and editing.

PD Kaiser: investigation, methodology, and writing—review and editing.

S Nueske: investigation, methodology, and writing—review and editing.

AM Scholz: investigation, methodology, and writing—review and editing.

E Schäffer: supervision, methodology, and writing—review and editing.

AJ Garcia-Saez: supervision, methodology, and writing—review and editing.

B Macek: supervision, methodology, and writing—review and editing.

U Rothbauer: conceptualization, formal analysis, supervision, funding acquisition, investigation, methodology, project administration, and writing—original draft, review, and editing.

### Conflict of Interest Statement

The authors declare that they have no conflict of interest.

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
