## [Reviewer comments · Life Science Alliance]

Life Science Alliance

Nanobodies as novel tools to monitor the mitochondrial fission factor Drp1

Theresa Froehlich, Andreas Jenner, Claudia Cavarischia-Rega, Funmilayo Fagbadebo, Yannic Lurz, Desiree Frecot, Philipp Kaiser, Stefan Nueske, Armin Scholz, Erik Schäffer, Ana Garcia-Saez, Boris Macek, and Ulrich Rothbauer

DOI: <https://doi.org/10.26508/lsa.202402608>

Corresponding author(s): Ulrich Rothbauer, University of Tübingen

Review Timeline:	Submission Date:	2024-01-22
	Editorial Decision:	2024-03-04
	Revision Received:	2024-05-07
	Editorial Decision:	2024-05-10
	Revision Received:	2024-05-14
	Accepted:	2024-05-15

Transaction Report:

March 4, 2024

Re: Life Science Alliance manuscript #LSA-2024-02608-T

Prof. Ulrich Rothbauer
University of Tübingen
Natural and Medical Science Institute
Auf der Morgenstelle 8
Reutlingen 72770
Germany

Dear Dr. Rothbauer,

Thank you for submitting your manuscript entitled "Nanobodies as novel tools to monitor the mitochondrial fission factor Drp1" to Life Science Alliance. The manuscript was assessed by expert reviewers, whose comments are appended to this letter. We invite you to submit a revised manuscript addressing the Reviewer comments.

Thank you for this interesting contribution to Life Science Alliance. We are looking forward to receiving your revised manuscript.

Sincerely,

B. MANUSCRIPT ORGANIZATION AND FORMATTING:

Reviewer #1 (Comments to the Authors (Required)):

In this study, the authors have identified two high-affinity nanobodies (Nbs) targeting Drp1, demonstrating their utility as (1) affinity agents for probing the Drp1 interactome, (2) labeling probes for advanced fluorescence microscopy, including super-resolution microscopy (SRM), and (3) chromobodies (Cbs) for live-cell visualization of Drp1 dynamics. The manuscript reads well and the findings should be of interest to readers of LSA. Overall, the manuscript supports its main claims. Nonetheless, certain aspects could be refined to enhance clarity and rigor.

1. The introduction lacks emphasis on the critical relevance of this study. A more compelling start would discuss the implications of Drp1 dysfunction in neurodegenerative diseases and cancer, positioning Drp1 as a pivotal biomarker. It would be beneficial to specify which methodologies and key discoveries from prior research underscore Drp1's linkage to these conditions.

2. Abbreviations in figures should be minimized or, if used, must be accompanied by their full denominations to ensure clarity.

3. Did the authors process ELISA for pre-bleed samples? The higher immune response of immunized bleed toward Drp1 than the negative control (BSA) doesn't mean the increase in immune response induced by Drp1. The pre-bleed sample is the best negative control. Please provide IC50 values.

4. α -Drp1 IgG was used as a positive control and comparison with D7 and D63 Nbs. Explain why this antibody was selected among others for comparison.

5. In Figure 3a, the authors provided SDS PAGE for HeLa cells but then provided a proteomics analysis for HEK cells (Figure 3b, and/or 3c). It is better to provide the data from the same cell line.

6. The authors claim the Nb-based affinity matrices facilitate rapid and efficient investigation of the Drp1 interactome across different conditions and tissues. However, proteomic analysis is only shown for HEK293 cells. Data supporting this claim for U2OS and HeLa cell lines should be provided.

7. "Fluorescence images showed cytosolic colocalization with GFP-Drp1 signal only for D7, whereas D63 showed no staining (Fig. S5). However, when we tested staining of endogenous Drp1, we did not observe any specific staining or colocalization with signals from a conventional Drp1 antibody (Fig. 4A)."

8. The manuscript mentions cytosolic colocalization with GFP-Drp1 for only one of the Nbs, leading to confusion regarding the specificity of staining in subsequent experiments. Clarification is needed on whether the experiments depicted in Figures S5 and 4A are identical and, if so, to improve the presentation for better clarity.

9. To confirm mitochondria fission was successfully induced by Carbonyl cyanide m-chlorophenylhydrazone (CCCP), please provide the data regarding the average length of mitochondria with and without CCCP treatment. In addition, include the time-lapse microscopy for HeLa cells without CCCP treatment as a critical negative control in Figure 7. This information will help verify that the stained mitochondria are truly undergoing the fission process.

Reviewer #2 (Comments to the Authors (Required)):

Froehlich et al puts forward an interesting study where nano bodies to monitor fission factor Drp1. It is indeed a good study more into a protocol angle than an explorative research. It's a valid idea to use STORM to real-time image Drp1. This approach could be applied to other similar proteins as well in future. In general, it's a good study.

A few minor comments are below

1. As done with in vivo detection of Drp1, what about use of such approach in visualizing other proteins with a similar context ?

2. This seems to be a good technique development helping in deciphering the role of Drp1 either by Co-IP, immunofluorescence or in vivo visualization.
3. In general, the visualization of Drp1 without interrupting its functional attributes might be of great use in studies related to model organisms.

Reviewer Comments

Nanobodies as novel tools to monitor the mitochondrial fission factor Drp1
Life Science Alliance manuscript #LSA-2024-02608-T

Reviewer #1 (Comments to the Authors (Required)):

In this study, the authors have identified two high-affinity nanobodies (Nbs) targeting Drp1, demonstrating their utility as (1) affinity agents for probing the Drp1 interactome, (2) labeling probes for advanced fluorescence microscopy, including super-resolution microscopy (SRM), and (3) chromobodies (Cbs) for live-cell visualization of Drp1 dynamics. The manuscript reads well and the findings should be of interest to readers of LSA.

We thank the reviewer for the positive response.

Overall, the manuscript supports its main claims. Nonetheless, certain aspects could be refined to enhance clarity and rigor.

1. The introduction lacks emphasis on the critical relevance of this study. A more compelling start would discuss the implications of Drp1 dysfunction in neurodegenerative diseases and cancer, positioning Drp1 as a pivotal biomarker. It would be beneficial to specify which methodologies and key discoveries from prior research underscore Drp1's linkage to these conditions.

We understand the reviewer's point of view. The manuscript as presented describes the development of Drp1-specific nanobodies and their multiple applications for the study of Drp1. However, in order not to overstate the content of the study, we have kept the role of Drp1 as a biomarker for monitoring pathophysiological processes rather brief in the original version. To follow the reviewer's suggestions, we have now addressed this point more clearly in the abstract and in the introduction of the revised manuscript and included appropriate references.

2. Abbreviations in figures should be minimized or, if used, must be accompanied by their full denominations to ensure clarity

We take note of this point and have revised our figure legends accordingly to include the full designations in the revised version of the manuscript.

3. Did the authors process ELISA for pre-bleed samples? The higher immune response of immunized bleed toward Drp1 than the negative control (BSA) doesn't mean the increase in immune response induced by Drp1. The pre-bleed sample is the best negative control. Please provide IC50 values (TF).

We agree with the reviewer and performed a serum ELISA with pre-immunization serum sample as control. The data (Figure 1, for reviewer only) show a significantly higher antibody response against Drp1 after immunization. However, due to the complex mixture of antibodies in the serum, determination of the IC50 provide only limited information, which is why we prefer not to indicate this in the figure.

Figure 1: Analysis of seroconversion upon vaccination with Drp1. A serum sample of the vaccinated alpaca (*Vicugna pacos*) was collected 13 weeks after the first dose. To test for Drp1-specific antibodies, a serum enzyme-linked immunosorbent assay (ELISA) was performed with the indicated dilutions of immunized (black) and non-immunized (blue) serum as negative control. Binding of Drp1-specific antibodies was detected by using an anti-heavy chain antibody conjugated to horseradish peroxidase. Notably, the data show the results from three technical replicates and include respective SDs, which are too small to be seen in the figure.

Most importantly, it has to be noted that the animal was previously immunized several times with other His-tagged proteins. In order to further exclude the determination of an antibody reaction against the His-tag, we now additionally included a non-specific His-tagged protein (Sortase-His₆) as a control, by which we could demonstrate the antibody response was specific for Drp1. The results of the serum ELISA using this novel control is now shown in the revised **Fig. S1A**. Finally, we would like to note that the serum ELISA is only a first indication of a successful immunization. The fact that we selected two high-affinity Drp1-binding Nbs from a comprehensive Nb gene library indicates, in our opinion, a positive immune response of the alpaca to immunization with Drp1.

4. α -Drp1 IgG was used as a positive control and comparison with D7 and D63 Nbs. Explain why this antibody was selected among others for comparison.

For our study, we preferred a monoclonal Drp1 antibody to avoid batch-to-batch variation, increase the reproducibility of the data and additionally reduce the possibility of binding competition between antibodies and our Nbs. The selected Drp1 monoclonal antibody (D6C7 from Cell Signaling) is already proven in various applications, including immunoblotting, immunoprecipitation and immunofluorescence, as evidenced by a large number (< 399) of citations.

5. In Figure 3a, the authors provided SDS PAGE for HeLa cells but then provided a proteomics analysis for HEK cells (Figure 3b, and/or 3c). It is better to provide the data from the same cell line.

We agree with the reviewer and have changed the figures shown in the revised manuscript accordingly. Western blot and proteomic analyses for HEK293 cells are now shown in **Figure 3** and the original Western blots of HeLa and U2OS cells are shown in a revised **Fig. S3**.

6. The authors claim the Nb-based affinity matrices facilitate rapid and efficient investigation of the Drp1 interactome across different conditions and tissues. However, proteomic analysis is only shown for HEK293 cells. Data supporting this claim for U2OS and HeLa cell lines should be provided.

We agree with the reviewer and performed additional experiments in which we analyzed the proteome of the two Drp1 nanotraps (D7 and D63) after immunoprecipitation from HeLa and U2OS cells. The new data are now summarized in the new **Fig. S5** and **Fig. S6**, and the additional findings are included in the results section and discussion of the revised manuscript.

7. "Fluorescence images showed cytosolic colocalization with GFP-Drp1 signal only for D7, whereas D63 showed no staining (Fig. S5). However, when we tested staining of endogenous Drp1, we did not observe any specific staining or colocalization with signals from a conventional Drp1 antibody (Fig. 4A)." The manuscript mentions cytosolic colocalization with GFP-Drp1 for only one of the Nbs, leading to confusion regarding the specificity of staining in subsequent experiments. Clarification is needed

on whether the experiments depicted in Figures S5 and 4A are identical and, if so, to improve the presentation for better clarity.

We acknowledge that these sentences, which are meant to describe the finding that only D7 (but not D63) recognizes exogenously expressed GFP-Drp1, while none of the monovalent Nbs are able to stain endogenous Drp1 in IF, are not easy to understand. Consequently we rephrased them accordingly in the revised version of the manuscript, which now reads as follows:

“First, we applied D7 and D63 as primary binding molecules in combination with a fluorescently labelled anti-VHH antibody in fixed and permeabilized U2OS cells transiently overexpressing GFP-Drp1. Only D7 showed colocalization with GFP-Drp1, whereas D63 showed no staining (**Fig. S7**), suggesting that this Nb does not bind Drp1 in IF. Further, when we tested IF for endogenous Drp1 in U2OS cells, we did not observe specific Drp1 staining for either Nb (**Fig. 4A**).”

We hope that this contributes to the comprehensibility of the data shown.

Notably, **Fig. S7** and **Fig. 4A** are not identical. **Fig. S7** shows U2OS cells transiently overexpressing GFP-Drp1, while **Fig. 4A** shows endogenous U2OS cells.

9. To confirm mitochondria fission was successfully induced by Carbonyl cyanide m-chlorophenylhydrazone (CCCP), please provide the data regarding the average length of mitochondria with and without CCCP treatment. In addition, include the time-lapse microscopy for HeLa cells without CCCP treatment as a critical negative control in Figure 7. This information will help verify that the stained mitochondria are truly undergoing the fission process.

We thank the reviewer for this comment. We used CCCP treatment as this has been previously described in the literature to efficiently induce mitochondrial fission and recruitment of Drp1 to mitochondria (see Voccoli et al, 2009 DOI: 10.1016/j.brainres.2008.11.026; Jones et al, 2017, DOI: 10.1007/s00018-016-2421-9; Park et al, 2018, DOI: 10.1016/j.toxlet.2017.12.004; Pascussi et al, 2021, DOI: 10.3390/ijms22137123). For clarification, we have mentioned this in the revised version of the manuscript and cited the relevant references.

To follow the advice of the reviewer, we additionally include novel data now showing a time-lapse imaging series of MitoTracker green stained HeLa cells without CCCP

treatment. This allows an optical assessment of the mitochondrial length over the same period of time as after treatment with CCCP (new **Fig. S12** compared to **Fig. 7**). **Figure 2** (for the reviewer only) also shows additional images of HeLa cells stained with the MitoTracker without and with CCCP treatment.

Figure 2: HeLa cells were stained with MitoTracker green and subsequently treated with or without 20 μ M cyanide m-chlorophenylhydrazine (CCCP). Representative confocal images were taken after 10 min, 70 min, and 130 min of treatment. Scale bar 25 μ m.

However, based on the small size and 3D- structure of mitochondria a quantification of mitochondrial fragments was not possible.

Reviewer #2 (Comments to the Authors (Required)):

Froehlich et al puts forward an interesting study where nanobodies to monitor fission factor Drp1. It is indeed a good study more into a protocol angle than an explorative research. It's a valid idea to use STORM to real-time image Drp1. This approach could be applied to other similar proteins as well in future. In general, it's a good study.

We thank the reviewer for the positive response.

A few minor comments are below:

1. As done with *in vivo* detection of Drp1, what about use of such approach in visualizing other proteins with a similar context ?

In the past, chromobodies have been applied to visualize a variety of targets, including cytoskeletal proteins (Schmidthals et al., 2010; Maier et al., 2015; Panza et al., 2015; Plessner et al., 2015), PCNA (Burgess et al., 2012) and beta-catenin (Dietrich et al., 2019; Traenkle et al., 2015) in cells but also in organisms. In the context of mitochondria, we have recently developed nano- and chromobodies against Miro1 (Fagbadebo et al., 2022), a protein associated with the outer membrane of mitochondria.

2. This seems to be a good technique development helping in deciphering the role of Drp1 either by Co-IP, immunofluorescence or *in vivo* visualization.

We thank the reviewer for the positive assessment of the herein demonstrated proof-of-principle applications of the presented Drp1 Nbs

3. In general, the visualization of Drp1 without interrupting its functional attributes might be of great use in studies related to model organisms.

We thank the reviewer for sharing our opinion on potential further applications of the presented Drp1-Nbs

May 10, 2024

RE: Life Science Alliance Manuscript #LSA-2024-02608-TR

Prof. Ulrich Rothbauer
University of Tübingen
Pharmaceutical Biotechnology
Auf der Morgenstelle 8
Reutlingen, Baden-Wuerttemberg 72770
Germany

Dear Dr. Rothbauer,

Thank you for submitting your revised manuscript entitled "Nanobodies as novel tools to monitor the mitochondrial fission factor Drp1". We would be happy to publish your paper in Life Science Alliance pending final revisions necessary to meet our formatting guidelines.

- please be sure that the authorship listing and order is correct
- please upload all figure files as individual ones, including the supplementary figure files; all figure legends should only appear in the main manuscript file
- please add the Twitter handle of your host institute/organization as well as your own or/and one of the authors in our system
- please add your main, supplementary figure, and table legends to the main manuscript text after the references section
- please add a Conflict of Interest statement to your main manuscript text
- please upload your Tables in editable .doc or Excel format
- please add callouts for Figures S3A-B; S5A-C, F, G; S6A-C, F, G; S10A-B; S14A-B; S15A-C and Table S1 to your main manuscript text
- please update the Data Availability statement with the Pride accession number once you receive it
- please incorporate the Supplemental References into the main Reference list

LSA now encourages authors to provide a 30-60 second video where the study is briefly explained. We will use these videos on social media to promote the published paper and the presenting author (for examples, see <https://docs.google.com/document/d/1-UWCfbE4pGcDdcgzcmiuJl2XMBJnxKYeqRvLLrLSo8s/edit?usp=sharing>). Corresponding or first-authors are welcome to submit the video. Please submit only one video per manuscript. The video can be emailed to contact@life-science-alliance.org

A. FINAL FILES:

B. MANUSCRIPT ORGANIZATION AND FORMATTING:

Sincerely,

May 15, 2024

RE: Life Science Alliance Manuscript #LSA-2024-02608-TRR

Prof. Ulrich Rothbauer
University of Tübingen
Pharmaceutical Biotechnology
Auf der Morgenstelle 8
Reutlingen, Baden-Wuerttemberg 72770
Germany

Dear Dr. Rothbauer,

Thank you for submitting your Methods entitled "Nanobodies as novel tools to monitor the mitochondrial fission factor Drp1". It is a pleasure to let you know that your manuscript is now accepted for publication in Life Science Alliance. Congratulations on this interesting work.

DISTRIBUTION OF MATERIALS:

Again, congratulations on a very nice paper. I hope you found the review process to be constructive and are pleased with how the manuscript was handled editorially. We look forward to future exciting submissions from your lab.

Sincerely,
